# Docetaxel for Breast Cancer Treatment-Side Effects on Ocular Surface, a Systematic Review

Elena Andreea Stoicescu [1,2,†], Marian Burcea [1,3,†], Raluca Claudia Iancu [1,2,*], Mirela Zivari [4], Alina Popa Cherecheanu [1,2], Inna Adriana Bujor [1], Cristina Rastoaca [2] and George Iancu [1,5]

1    Faculty of Medicine, "Carol Davila" University of Medicine and Pharmacy, 37 Dinisie Lupu Street, 020021 Bucharest, Romania; ely_anghel@yahoo.com (E.A.S.); mnburcea@gmail.com (M.B.); alina_cherecheanu@yahoo.com (A.P.C.); moia_innutza@yahoo.com (I.A.B.); klee_ro@yahoo.com (G.I.)
2    Department of Ophthalmology, Emergency University Hospital, 169 Independentei Street, 050098 Bucharest, Romania; cristina.rastoaca@gmail.com
3    Department of Ophthalmology, Emergency Ophtalmology Hospital, 1 Lahovari Square, 030167 Bucharest, Romania
4    Department of Psychology, Emergency University Hospital, 169 Independentei Street, 050098 Bucharest, Romania; mirelaziv@yahoo.com
5    Filantropia Clinical Hospital of Obstetrics and Gynecology, 11-13 Ion Mihalache Bvd, 011132 Bucharest, Romania
*    Correspondence: raluca.iancu@umfcd.ro
†    Contributed equally.

**Abstract:** Docetaxel is a very effective chemotherapeutic agent for the treatment of metastatic or locally advanced breast cancer. Epiphora (hyperlacrimation) has been shown to be the most common eye condition in patients receiving docetaxel-based chemotherapy. This symptom does not decrease visual acuity, but decreases the quality of life. Daily activities (reading, working on the computer, watching TV, and so on) are affected, with patients complaining about an alteration of daily life with the appearance of this symptom. The mechanism by which epiphora occurs is considered to be the canalicular stenosis, but the trials on the subject failed to reach statistical significance. The objective of this scoping review is to determine whether there is a treatment regimen-dependent relationship between docetaxel administration and the presence of epiphora in women with breast cancer. The inclusion criteria were met by 10 trials, from which one was excluded owing to data selection biases. Accordingly, nine studies were evaluated quantitatively and qualitatively in the present review. We included subjects with docetaxel as single treatment or docetaxel in combination with other chemotherapy compounds. The occurrence of epiphora among subjects treated with docetaxel, regardless of the therapeutic regimen used, was statistically significant ($p = 0.005$). The proportion of patients with epiphora after weekly administration of docetaxel (54 out of 131 subjects, 41.22%) was different compared with that of those who received docetaxel at three week intervals (112 out of 325 subjects, 34.15%), but the difference between the two was not statistically significant ($p = 0.732$). The present study demonstrates that epiphora occurs more frequently in patients receiving weekly docetaxel-based chemotherapy than those taking the three-weekly regimen, but the difference is not statistically significant. Ophthalmologic assessment of all patients starting this treatment is recommended. The causal relationship between canalicular stenosis and epiphora is not fully elucidated as long as this ocular symptom occurs in women who do not have stenosis of the lacrimal system. Further well-designed trials are required to bring new insights into the mechanisms of epiphora pathogenesis in subjects treated with docetaxel.

**Keywords:** docetaxel; epiphora; hyperlacrimation; breast cancer

## 1. Introduction

The ocular surface, the tear film, the lacrimal glands, and the eyelids represent a functional unit. Any changes in the homeostasis of this unit lead to the appearance of

pathological manifestations in the ocular surface [1]. A variety of chemotherapeutic agents have been reported to cause epiphora, like 5-Fluorouracil, docetaxel, or mitomycin-C [2].

Of the side effects in women with breast cancer who are treated with docetaxel, the most common is epiphora. It decreases quality of life, not visual acuity, as shown by most studies [3–6].

The purpose of this review is to determine whether there is a dependent relationship between docetaxel administration in women with breast cancer and the occurrence of epiphora in them.

Docetaxel is an antineoplastic agent that acts by disorganizing the microtubular network of cells [3,4], which is essential for vital cellular functions during mitosis and interphase (antimitotic). Thus, docetaxel favors the assembly of tubulin into stable microtubules and inhibits their disassembly, decreasing the level of free tubulin. Its use in breast cancer treatment regimen both as single chemotherapeutic agent or in combination with other chemotherapeutic agents has led to improved patient outcomes [3,5]. It is administered by intravenous infusion for one hour, once every 3 weeks (dose between 60 and 100 mg/m$^2$) or weekly (25 and 35 mg/m$^2$) [6].

The safety profile of docetaxel depends on the dose and the rate of administration [6]. Therapeutic indications are as follows: breast cancer as monotherapy or in combination with another agent (docetaxel + doxorubicin + cyclophosphamide; docetaxel + trastuzumab; docetaxel + capecitabine) [3,6,7]; bronchopulmonary cancer other than small cell cancer (as monotherapy or in combination) [3,8]; hormone-resistant or metastatic prostate cancer (in combination) [3,9]; gastric adenocarcinoma (in combination) [3,10]; and head and neck cancer.

Weekly docetaxel administration at a dose of 20–42 mg/m$^2$ [3,11] is more commonly used in metastatic disease, where it has been shown to have fewer neutropenic complications and increased patient tolerance owing to a reduced dose range [12,13] compared with the three-week treatment regimen [3,11,14]. If the weekly docetaxel dose exceeds 42 mg/m$^2$/week, then the risk of febrile neutropenia increases [11–14].

The kinetic profile of the drug does not depend on the dose having a three-compartment pharmacokinetic model with a half-life of 4 min, 36 min, and 11.1 h (late phase is explained by the slow efflux of docetaxel from the peripheral compartment) [15]. It binds to 95% plasma proteins and is excreted in the urine and feces for 7 days [15].

Docetaxel is an antineoplastic agent with many side effects: haematological (e.g., febrile neutropenia [3,12], neutropenic infection), gastrointestinal (e.g., enterocolitis) [3,16], hypersensitivity reactions (e.g., bronchospasm, severe skin reactions [3,17], severe hypotension), fluid retention [3,17], respiratory reactions [3,18] (e.g., acute respiratory distress syndrome, interstitial pneumonia, respiratory failure), cardiac reactions [3] (heart failure, ventricular tachycardia), ocular reactions (epiphora [3,6,11,14], cystoid macular edema [3,19], optic neurotoxicity [3,20], erosive conjunctivitis [21], keratopathy), and nervous system disorders (e.g., sensory and motor peripheral neuropathy) [3,22].

In women with breast cancer, it is recommended that glucocorticoids should be administered orally, 16 mg/day, as premedication for 3 days before starting treatment with docetaxel to reduce the toxicity of the drug (decreases the incidence and severity of fluid retention, as well as hypersensitivity reactions) [3,23].

At the ocular level, the epiphora is the most common ocular adverse event associated with docetaxel therapy [3,24,25]. Epiphora is due to excessive secretion and/or obstruction in tear drainage. With blinking, the tears are pushed medially, reaching the lacrimal points into the lacrimal canal, then into the common lacrimal duct, then in the lacrimal sac, then flowing into the inferior nasal meatus [11]. Tear drainage maintains its direction owing to the valves of the drainage system, thus not allowing tears' backflow [11].

The reduced tear flow may be the result of inflammation, infections, strictures, tumors encountered along the drainage system, or eyelid dysfunction [11]. The permeability of the drainage channels is checked by sounding and irrigating them, introducing a probe through the tear point to the medial wall of the lacrimal sac [11].

## 2. Materials and Methods

The objective of this scoping review is to determine whether there is a treatment regimen-dependent relationship between docetaxel administration and the presence of epiphora in women with breast cancer. We included subjects with docetaxel as single treatment or docetaxel in combination with other chemotherapy compounds.

*Search strategy.* A thorough search of the medical databases was conducted in order to identify the original research papers published between January 2001 and November 2020. The following databases were searched: Medline/PubMed, Embase, Cochrane, and Web of Knowledge. The following keywords were used: "docetaxel" and "epiphora" or "ocular surface" or "lacrimal system" or "lacrimation" or "watering eyes".

*Inclusion criteria.* We included studies in which the participants had breast cancer and treatment with docetaxel as a single agent or docetaxel in combination with other chemotherapy regimen. Other chemotherapy compounds used in association with docetaxel were as follows: trastuzumab, doxorubicin, cyclophosphamide, fluorouracil, and epirubicin. The treatment regimens selected for inclusion were weekly administration and the administration at every 3 weeks.

*Data collection, data analysis, and outcomes.* The study was conducted and reported according to Population or Problem, Intervention or Exposure, Comparison, and Outcome (PICO) guidelines [26]. The data collection was performed as follows: one reviewer extracted data from the selected studies and two reviewers checked these data for accuracy. Two review authors independently evaluated the titles and abstracts. For each trial, the authors recorded participants' data using a standard data extraction method to record data on patients' characteristics, number of participants, treatment regimens, primary outcome, and secondary outcomes. The primary outcome included the appearance of epiphora after treatment initiation. Secondary outcomes were docetaxel dose intensity and cumulative dose. Because of the fact that not all the selected trials included values for cumulative dose, this particular parameter was included only for descriptive statistics.

*Statistical analysis* was performed using SPSS 26 software platform (IBM). Analysis of variance (ANOVA), bivariate correlation analysis, and Mann–Whitney U Test were performed, along with descriptive statistics. A $p$-value of 0.05 or lower was considered statistically significant.

## 3. Results

The medical databases search resulted in the identification of 53 articles. From those, 45 were selected for abstract review. After abstract review, we selected 34 articles for full-text review. The inclusion criteria were met by 10 trials, from which one was excluded owinf to data selection biases. Accordingly, nine studies [7,14,25,27–32] were evaluated quantitatively and qualitatively in our review (Table 1). From the eligible trials, a total of 456 women were included in the analysis.

*Methodology of studies.* Study design was not homogeneous across different studies. The selected trials were as follows: randomized controlled studies (two) and observational studies, prospective (three) and retrospective (three). We also included one case control study. Overall, two treatment regimens of docetaxel administration were identified: weekly administration and administration every 3 weeks. All trials measured our primary outcome. Four of the trials selected [7,14,28,29] included subjects treated with both regimens, weekly and every 3 weeks docetaxel; the rest of the trials [25,27,30–32] comprised women treated with a single regimen.

The mean age of the subjects included in the analysis was similar in all selected trials. The presence of epiphora among subjects treated with docetaxel, regardless of the therapeutic regimen used, was statistically significant ($p$ = 0.005). The proportion of patients with epiphora after weekly administration (54 out of 131 subjects, 41.22%) was different compared with that of those who received docetaxel at every three weeks (112 out of 325 subjects, 34.15%), but the difference between the two was not statistically significant ($p$ = 0.732). The mean number of patients with epiphora in weekly administration

group was 9, and in the every 3 weeks administration group, it was 12.44 (Figure 1). In Figure 2, there is a comparison between the rate of epiphora recorded in each trial and the administration regimen used.

**Table 1.** The selected clinical trials and the specific chemotherapeutic regimens.

| Study | Study Type | Administration Regimen | Total No. of Subjects | Epiphora |
|---|---|---|---|---|
| Noguchi Y 2019 [27] | Case control | Every 3 weeks TH, TC, TCH | 89 | 7 |
| Leyssens B 2009 [30] | RCT | Weekly | 20 | 9 |
| Esmaeli B 2002 [14] | Retrospective | Weekly TH | 18 | 14 |
| Esmaeli B 2002 [14] | Retrospective | Every 3 weeks TA | 18 | 2 |
| Esmaeli B 2001 [25] | Observational | Weekly T | 3 | 3 |
| Tabernero J 2004 [29] | RCT | Weekly T | 41 | 3 |
| Tabernero J 2004 [29] | RCT | Every 3 weeks T | 42 | 0 |
| Tsalic M 2006 [31] | Prospective | Weekly T | 21 | 7 |
| Esmaeli B 2006 [7] | Prospective | Weekly T | 28 | 18 |
| Esmaeli B 2006 [7] | Prospective | Every 3 weeks T | 28 | 11 |
| Noguchi Y 2016 [32] | Retrospective | Every 3 weeks T | 48 | 6 |
| Chan A 2013 [28] | Prospective | Every 3 weeks FEC-T | 14 | 14 |
| Chan A 2013 [28] | Prospective | Every 3 weeks TCH | 17 | 15 |
| Chan A 2013 [28] | Prospective | Every 3 weeks TAC | 55 | 47 |
| Chan A 2013 [28] | Prospective | Every 3 weeks TC +/− H | 14 | 9 |

T = docetaxel, H = trastuzumab, C = cyclophosphamide, A = doxorubicin, F = fluorouracil, E = epirubicin.

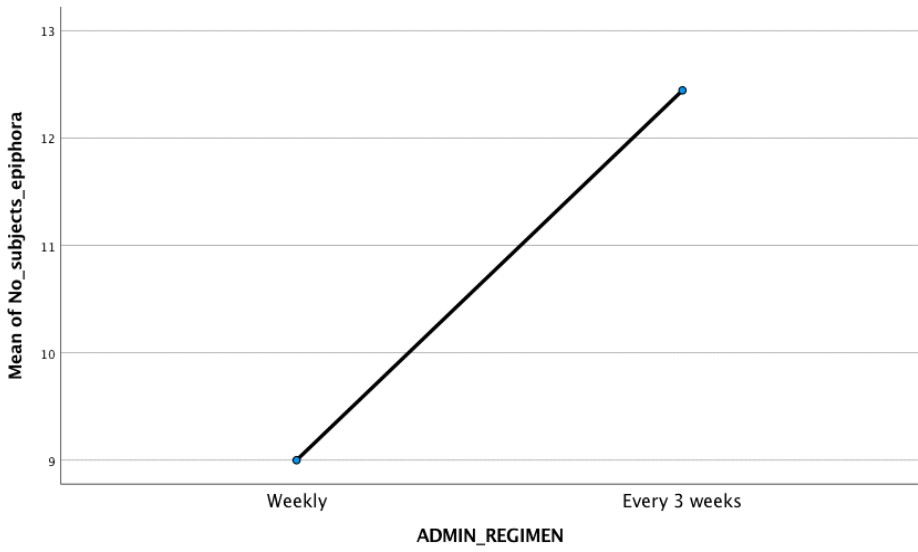

**Figure 1.** The mean number of subjects with epiphora in the two administration regimens (9 for weekly administration, 12.4 for every 3 weeks regimen).

The mean dose intensity for patients treated by weekly administration was 33.36 mg/m$^2$/week (SD = 2.53). The mean dose intensity for patients treated by every 3 weeks administration was 26.11 mg/m$^2$/week (SD = 4.37). The difference between dose intensity for the therapeutic regimen is statistically significant ($p < 0.0001$), and there is a negative correlation between the dose intensity value and the number of weeks between doses. The median for dose intensity independent of the treatment regimen was 30 mg/m$^2$/week, and the incidence of the epiphora was similar (not statistically significant different) for doses over and under this value (Figures 3 and 4).

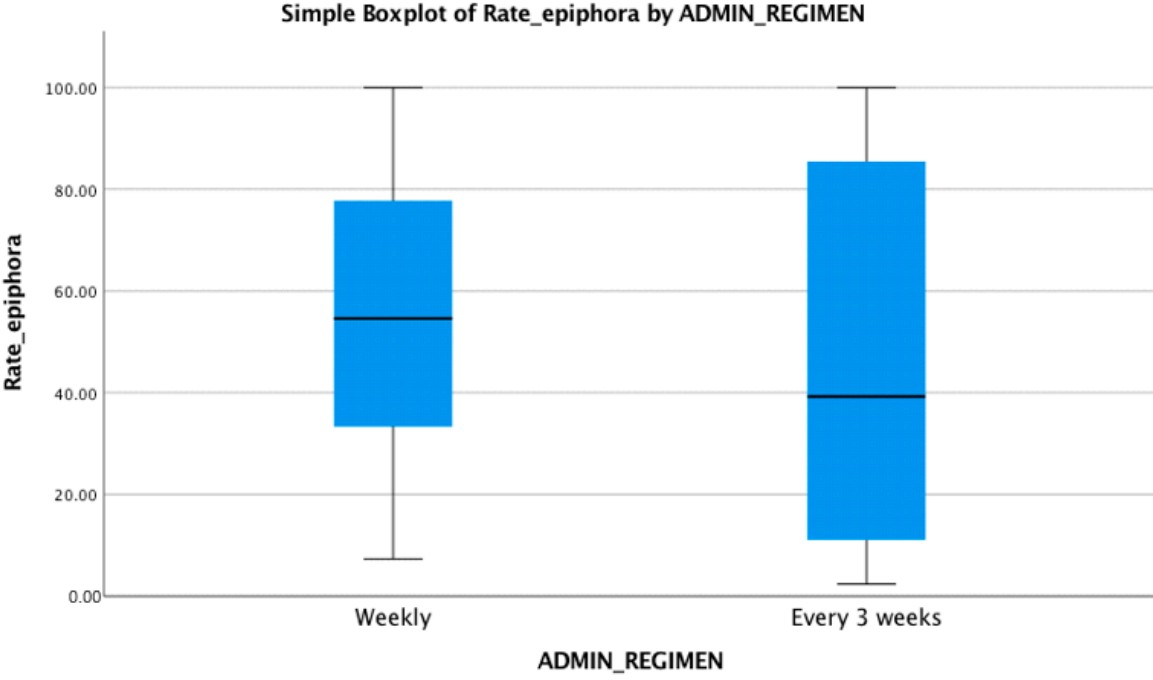

**Figure 2.** Rate of epiphora recorded in each administration regimen used, as median (54.64 for weekly regimen and 39.28 for every 3 weeks administration), maximum, and minimum values.

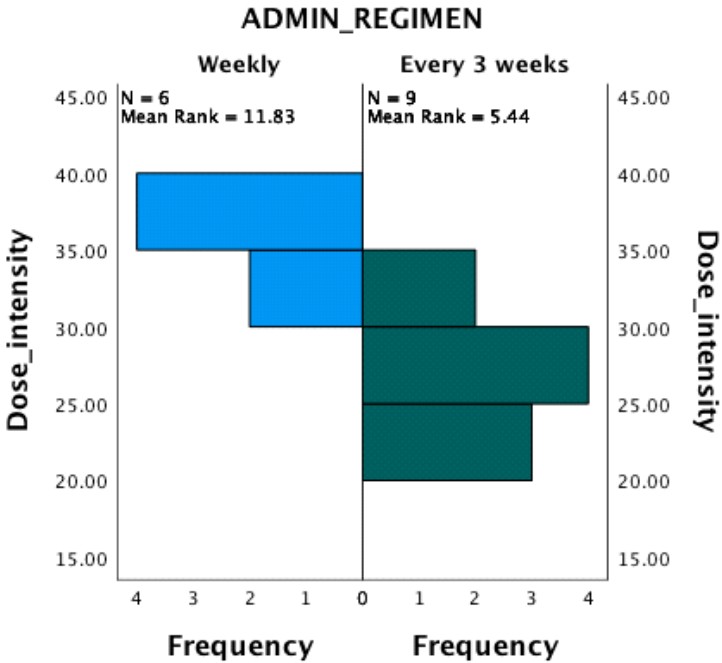

**Figure 3.** The number of subjects with epiphora is similar for dose intensity lower and greater than 30 mg/m$^2$/week.

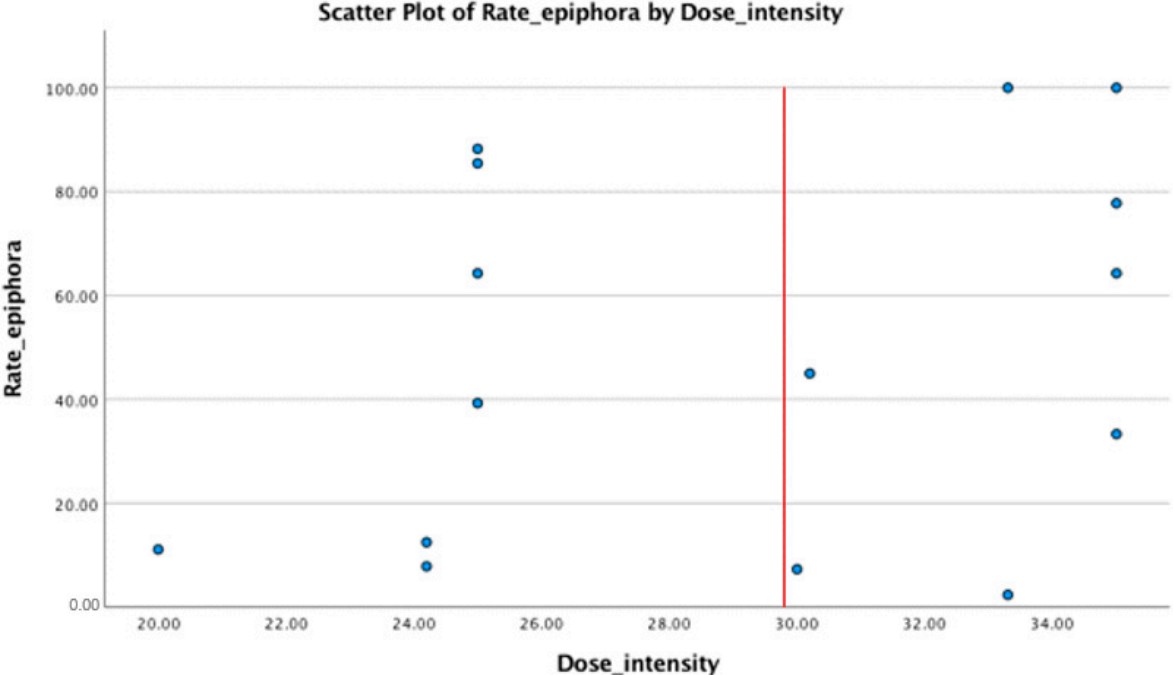

**Figure 4.** The distribution of rate of epiphora by dose intensity is approximately equal related to the red line. The red line represents the median value of the dose intensity (=30 mg/m$^2$/week).

## 4. Discussion

Docetaxel is a very effective chemotherapeutic agent for the treatment of metastatic or locally advanced breast cancer [5,33]. Life expectancy increased with its introduction in breast cancer treatment [5,34]. The results of three randomized phase III studies showed a significant improvement in life expectancy in docetaxel-treated patients compared with anthracycline-treated patients [34–36]. At the ocular surface level, the epiphora is the most common adverse reaction [3,11,24,33].

The most frequently used chemotherapy regimen in breast cancer is intravenous administration of docetaxel at 3-week intervals, with doses between 20 and 33.3 mg/m$^2$/week [3,37]. The aim of this scoping review is to determine if there is a treatment regimen-dependent relationship between docetaxel administration and the presence of epiphora in women with metastatic or non-metastatic breast cancer.

All trials included in this review reported the presence of epiphora after docetaxel administration in women with breast cancer. There were nine trials included in this review. Four of them included subjects with both treatment regimens (weekly and every 3 weeks). The presence of the epiphora was different between the two treatment groups, but the difference was not clinically significant ($p = 0.732$). In all included subjects, the symptoms were relatively mild, and quite similar among participants. Among included studies, the only statistically significant difference was related to the dose intensity of the docetaxel treatment and the number of weeks between administrations; there was a negative correlation ($p < 0.0001$).

In six out of nine trials, the cumulative dose was included. The median value of the cumulative dose was 291.0 (42.3–937.4). In previous trials, the multivariate regression analysis revealed that a higher cumulative dose was significantly associated with the presence of epiphora [14,27], and that a higher dose was significantly associated with the canalicular stenosis [14].

Thus, it is clear that weekly administration of this drug leads to a more frequent occurrence of epiphora than its administration at three weeks [7,14,38]. Burstein and colleagues showed in a phase II study that epiphora was present in 50% of women who

received docetaxel weekly [39]. Epiphora usually occurs within 12–16 weeks of starting docetaxel treatment [39].

The mechanism by which epiphora occurs is shown in several studies as canalicular stenosis [7,20,38,40], but it is not statistically significant. Once epiphora occurs, it affects daily activities (reading, computer work, driving, and so on). Docetaxel is present in the tear film [41,42] and its permanent contact with the mucosa of the nasolacrimal duct and the lacrimal ducts favors the appearance of inflammation $+/-$ fibrosis at this level, leading to their narrowing/stenosis [33,40]. The cumulative dose of docetaxel in patients with severe canalicular stenosis was higher than in patients with moderate canalicular stenosis, although the difference was not statistically significant [40]. Thus, patients with advanced disease who must receive this chemotherapy regimen weekly for longer periods of time have a higher risk of developing canalicular stenosis than patients taking short courses.

A randomized phase II study shows that conservative treatment of canalicular stenosis with artificial tears and/or topical steroids is not effective in patients receiving docetaxel weekly.

For patients receiving weekly docetaxel-based chemotherapy, treatment of the epiphora has been indicated by implanting temporary silicone tubes in the canaliculi [43] as early as the beginning of treatment [25], especially if the patient begins to become symptomatic. In order to avoid a severe stenosis or even their closure, this moment requires surgery such as dacryocistorinostomy (DCR) or conjunctival dacryocistorinostomy (conjunctival DCR) with the implantation of a Pyrex glass tube [24,33,40,44]. In patients receiving docetaxel-based chemotherapy once every 3 weeks, previous studies recommend monitoring every 6 weeks with probing and irrigation of the tear system and short-term treatment of topical steroids [33]. If patients are treated every 3 weeks for a long period of time, and the epiphora persists, it is also recommended to implant temporary silicone tubes at the level of the tear ducts [33,43] Silicone tubes remain implanted throughout the treatment and even 4–6 weeks after the end of the chemotherapy treatment.

However, there are studies showing that epiphora is also present in patients without canalicular stenosis, receiving weekly docetaxel-based chemotherapy [28]. Drainage of the lacrimal system was checked before, during, and at the end of treatment by computed tomography (computed tomography dacryocystography: CT-DCG) [28]. There are also studies showing that, at 4 months after the last docetaxel administration, almost 70% of patients no longer manifest epiphora, 29% report intermittent tearing, and 1% still report epiphora [28].

Weekly docetaxel administration has been shown to have fewer myelodepressive reactions than administration at three weeks (fewer neutropenic complications and increased patient tolerance, by reducing the dose range and maintaining therapeutic efficacy with better survival) [12,13]. However, the weekly dose of docetaxel should not exceed 42 mg/m$^2$ because dose-limiting side effects such as febrile neutropenia occur [11]. The best systemic tolerability for the weekly administration was observed at a dosage of 30–36 mg/m$^2$ [11].

Ideally, all patients starting this treatment should be evaluated ophthalmologically before the treatment initiation, then monthly (including visual acuity, Shirmer test, tear osmolarity, tear time of the non-invasive tear film, and meibogram; the biomicroscope will follow up the appearance and position of the tear point and check the permeability of the canaliculi and the nasolacrimal duct). Thus, once this ocular adverse reaction is discovered, the ophthalmologist will manage this adverse effect so that the patient's quality of life is not altered. This can only happen if there is a good collaboration between the oncologist and the ophthalmologist, so that as soon as this symptom occurs, the oncologist should send the patient to an ophthalmological consultation so that he can properly manage this side effect, and thus the quality of daily life of patients can remain unaffected.

One must also keep in mind that, in the vast majority of patients, the epiphora recovers shortly after the end of docetaxel-based chemotherapy.

The limitations of the paper are represented by the low number of clinical trials found in the literature and their nonrandomized design.

## 5. Conclusions

The present study demonstrates that epiphora occurs more frequently in patients receiving weekly docetaxel-based chemotherapy than those taking the three-weekly treatment, but the difference is not statistically significant. Ophthalmologic assessment of all patients starting this treatment is recommended.

Although most docetaxel-treated patients believe that epiphora is a self-limiting problem that resolves a few weeks after stopping docetaxel chemotherapy, it is particularly common and severe in patients receiving docetaxel weekly, affecting quality of life.

It is very important to have a good collaboration between the oncologist and the ophthalmologist durring the treatment with docetaxel in women with breast cancer, in order to manage in time this ocular symptom, which does not decrease the visual acuity, but decreases the quality of life.

A monthly visit to the ophthalmologist is recommended as soon as patients have started docetaxel-based chemotherapy.

The causal relationship between canalicular stenosis and epiphora is not fully elucidated as long as this ocular symptom occurs in women who do not have stenosis of the lacrimal system. Certainly, future prospective studies will establish the basic mechanism that leads to epiphora in patients undergoing docetaxel-based chemotherapy.

**Author Contributions:** E.A.S., R.C.I. and G.I. designed the article. I.A.B., C.R., A.P.C. and M.Z., M.B., searched the databases and reviewed the literature. E.A.S. and R.C.I. wrote the first draft of the manuscript. G.I. critically revised the manuscript. All authors have read and agreed to the published version of the manuscript.

**Funding:** The research received no external funding.

**Institutional Review Board Statement:** No institutional review was required for this review paper.

**Informed Consent Statement:** No informed consent statement was required for this review.

**Data Availability Statement:** There are no original data to make available for this review paper.

**Conflicts of Interest:** The authors declare that they have no conflict of interest.

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
