# Peer review of "Docetaxel for Breast Cancer Treatment-Side Effects on Ocular Surface, a Systematic Review"

_processes, doi:10.3390/pr9071086_

Round 1

Reviewer 1 Report

The manuscript presents an interesting review study.

But some concerns shall be adressed:

In the introduction is missing the aim of realisation of such study. Please can you just explain a bit more the novelty of this review and why now.

The objective is shortly listed in M&M, why?

Please change the format of Table I, it is a bit wiread in the present form.

What about the age of the women included in this study analysis?

Conclusions need to be improved.

Author Response

The purpose of the review is to determine whether the presence of epiphora in women with breast cancer treated with docetaxel îs dependent on treatment regim:weekly or 3 week course. 

I redid the table 1.

In the published and studies studies for the review, emphasis was placed on the treatment regim of these patients, age not Boeing a criterion.

I will Take all your advice into account.

Thanks for your advice. 

Reviewer 2 Report

Almost all drugs used in cancer chemotherapy have some kind of side effects. Therefore, the practicing oncologist should be aware of such complications. This review contains materials about such complications in the treatment of breast cancer with mono- or complex use of decitaxel. The review will be of interest to oncologists to understand the complications of cancer therapy and predict the consequences of the proposed therapy. The manuscript can be accepted for publication.

312

  1. Bita Esmaeli ,Gabriel N Hortobagyi , Francesco J Esteva , Daniel Booser,M Amir Ahmadi,Edgardo Rivera,Rebecca Ar- buckle,Ebrahim Delpassand, 313

Laura Guerra,Vicento Valero :Canalicular stenosis secondary to weekly versus every-3-weeks docetaxel in patients with metastatic 314 breast cancer. Ophthalmology 109(6):1188–1191,2002

Have to be

  1. Bita Esmaeli ,Gabriel N Hortobagyi , Francesco J Esteva , Daniel Booser,M Amir Ahmadi,Edgardo Rivera,Rebecca Ar- buckle,Ebrahim Delpassand, Laura Guerra,Vicento Valero :Canalicular stenosis secondary to weekly versus every-3-weeks docetaxel in patients with metastatic breast cancer. Ophthalmology 109(6):1188–1191,2002

Author Response

Bună ziua. 

MulÈ›umesc pentru comentariul dumneavoastră. 

Am ținut cont și am revizuit.

Reviewer 3 Report

The authors have created an extrem well written systematic review regarding epiphora after docetaxel therapy.

Nevertheless the authors have to explore a bit the importance of the side effect epiphora. The clinical relevance of epiphora is not by itself clear to the audience and has to be addressed in the abstract, the introduction and the discussion. Are there any studies regarding quality of life or regarding visual acuity reduction etc.? 

Further,  I would recommend to orientate at PICO for the methods  (e.g. https://libguides.murdoch.edu.au/systematic/PICO) and then name the review a systematic review (also in the title). Almost all steps therefore have been done, but just have to be addressed in the methods.

In the discussion, I am missing a limitation section.

Finally, the message at the end is weak. Is there any recommondation for the clinical day? Are both docetaxel regimes equal for cancer treatment? This has to be addressed in the results, discussion and conclusion

Author Response

Almost all studies say that there îs a decrease in patients quality of life, not necessarily in vizual acuity.In clinical practicepatients quality of life, not necessarily in vizual acuity.In clinical practice,

there must be a very good  collaboration between the oncologist and the ophthalmologist, as soon as the  patient presents with epiphora, the oncologist must send her immediately to an ophthalmological consultation. 

Thanks for your advice. I will take them into account. 

Reviewer 4 Report

Manuscript entitled "DOCETAXEL FOR BREAST CANCER TREATMENT – OCULAR SIDE EFFECTS ON OCULAR SURFACE"

This is a useful and clinically relevant work describing the ocular side effect of docetaxel. I personally suggest it is almost acceptable in the present format. It would be improved by adding the review on this topic in discussion in more detail.

Author Response

Buna ziua.

Va multumesc pentru aprecieri si interesul pentru articol.

Voi tine cont de sfaturile date.

Cu respect,dr.Stoicescu Elena

Round 2

Reviewer 1 Report

The authors did not improve th manuscript adressing the reviewers suggestions/recommandation/advices.

The Table 1 has been changed, but it is the only re-considered by the authors.

Author Response

Thank you for you recommendations and for you support,I appreciate.

I will answer you point by point :

-in the Abstract the purpose of this review is mentioned ( "The objective of this scoping review is to determine whether there is a treatment regimen dependent relationship between docetaxel administration and the presence of epiphora in women with breast cancer.") but I will mention this in the Introduction as well as you advised me.Thank you.

-I do not understend exactly the question,, The objective is shortly listed in M&M,why?" , do you want be more explicit,please?Thank you !

-I redid the table 1.Thank you

-my article is a systematic review,not a study.In all articles found and read for this review the age of the patients was not mentioned and I suspect that this happened because their purpose was to show the dependence relationship between docetaxel treatment in women with breast cancer and the appeareance of epiphora.Thanky you

-I improved the conclusions,as you recommended.Thank you.

-

Reviewer 3 Report

So fare, I do see, how my raised point have been answered and how the manuscript has been changed therefore (not much work at all).

The clinical relevance has not been described so fare. At least a limitation section has to be included. I do not see any valuable conclusion.

I seems, that, basically, the authors do not care at all.

Author Response

Thank you for you advice and support. I apologize for not understanding exactly what you asked for, I am now trying to answer you point by point. Thanks for understanding .

-almost all the articles read on this topic, which are mentioned in my review, have described that once the epiphora appears, the quality of life of patients treated with docetaxel decreases,not necessarily visual acuity.I improved this idea in my article by adding in the Abstract, Introduction and Discussions the lines 22,23,24,25,60,61,62,63,64,65,272,273,274,275,276.

All the patients I have studied so far have complained about this embarrassment, this excessive leaking of tears on the cheeks, and not a decrease in visual acuity.

-I changed the title a bit, as you recommended : "Docetaxel for breast cancer treatment-Side effects on ocular surface,a systematic review "

-I used PICO for Materials and Methods, as you recommended (lines 131,132,367,368)

-I also took into account your advice and added to the Discussions the limitations of this paper (lines 279,280)

-I improved the conclusions.(lines 284,285,286,287,288,289,290,291,292)

In Results, Discussions and Conclusions it is specified that the weekly docetaxel regimen leads to a more frequent occurrence of epiphora than the 3-week treatment regimen.(lines 167,168,169,170,220,221,222,223,224,230,231,232,233,234,235,259,260,261,262,280,281,282,283).

The aim of this scoping review is to determine if there is a treatment regimen dependent relationship between docetaxel administration and the presence of epiphora in women with metastatic or non-metastatic breast cancer

Thank you once again for your understanding and support.

King Regards ,dr Stoicescu Elena